# Concomitant Use of Elexacaftor/Tezacaftor/Ivacaftor and Etanercept in a Cystic Fibrosis Patient with Juvenile Idiopathic Arthritis

**DOI:** 10.3390/jcm12051730

**Published:** 2023-02-21

**Authors:** Sara Immacolata Orsini, Edoardo Marrani, Ilaria Pagnini, Giovanni Taccetti, Vito Terlizzi, Gabriele Simonini

**Affiliations:** 1Department of Woman, Child and of General and Specialized Surgery, Università degli Studi della Campania “Luigi Vanvitelli”, 80138 Naples, Italy; 2Paediatric Rheumatology Unit, Meyer Children’s Hospital IRCCS, 50139 Florence, Italy; 3Department of Paediatric Medicine, Meyer Children’s Hospital IRCCS, Cystic Fibrosis Regional Reference Centre, 50139 Florence, Italy

**Keywords:** pediatric rheumatology, juvenile idiopathic arthritis, cystic fibrosis

## Abstract

Patients with cystic fibrosis often complain of joint manifestations. However, only a few studies have reported the association between cystic fibrosis and juvenile idiopathic arthritis and addressed the therapeutic challenges of these patients. We describe the first paediatric case of a patient affected by cystic fibrosis, Basedow’s disease and juvenile idiopathic arthritis who was contemporarily treated with elexacaftor/tezacaftor/ivacaftor (ELX/TEZ/IVA) and anti-tumor necrosis factor α (anti-TNFα). This report seems to reassure regarding the potential side effects of these associations. Moreover, our experience suggests that anti-TNFα is an effective option in CF patients affected by juvenile idiopathic arthritis, and is even safe for children receiving a triple *CFTR* modulator.

## 1. Introduction

Patients with cystic fibrosis (CF) often complain of joint manifestations., mostly episodic arthritis, pulmonary hypertrophic osteo-arthropathy (HPOA), antibiotic-induced pain and spinal pain [1].

Many systemic diseases have been reported in association with CF, including rheumatoid arthritis (RA) and sarcoidosis [1]. However, only few studies have reported the association between CF and juvenile idiopathic arthritis (JIA) and addressed the therapeutic challenges of these patients [1].

We describe the first pediatric case of a patient affected by CF, Basedow’s disease and JIA who was contemporarily treated with elexacaftor/tezacaftor/ivacaftor (ELX/TEZ/IVA) CF transmembrane conductance regulator (CFTR) modulator, methimazole and anti-tumor necrosis factor α (anti-TNFα).

## 2. Results

A fifteen-year-old Caucasian girl diagnosed with CF with pancreatic insufficiency by positive neonatal screening (genetic profile: F508del/F508del, sweat chloride (SC): 90–86 mEq/L) and treated at CF Regional Centre of Florence, Italy, according to CF standard of care [2]. The CF was characterized by mild lung disease with Forced Expiratory Volume in the first second (FEV1) in the range of 78–83% in 2021; there was no need for antibiotic therapy, and segmental bilateral bronchiectasis was limited to the upper and lower lobes at chest computer tomography. A throat swab culture showed the presence of Methicillin Sensitive Staphylococcus Aureus. No related CF diseases were found such as hypochloremic metabolic alkalosis, acute pancreatitis or CF related diabetes. Furthermore, she was also affected by Basedow’s disease with hyperthyroidism. The autoimmune profile showed the presence of anti-TSH-receptor auto-antibodies, anti-thyroid peroxidase auto-antibodies and antithyroglobulin auto-antibodies.

A thyroid ultrasound at the diagnosis showed the increased size, inhomogeneous echo structure, increased vascularization and pseudo nodular aspect of the left lobe.

Methimazole was then added from the age of 11 years achieving good disease control. The lumacaftor (LUM)-IVA therapy was started at 12 years, according to Italian legislative directives. No improvement was observed in the main clinical outcomes (such as FEV1 or BMI) or SC testing.

At the age of 15, she developed polyarticular arthritis, with involvement of all the metacarpophalangeal and proximal interphalangeal joints, wrists, right elbow and left ankle. The girl presented a morning stiffness lasting several hours, with no systemic symptoms or skin manifestations.

Inflammatory markers were in the normal range, antinuclear antibodies (ANA) were positive (1:320; speckled pattern), as were the rheumatoid factor (RF) (43 UI/mL with n.v. < 15) and anti-cyclic citrullinated peptide antibodies (anti-CCP) (214 UI/mL with n.v. < 20). Complement fraction C3 and C4 were in the normal range.

Extractable nuclear antigen antibodies and anti-double stranded DNA antibodies were negative. Ultrasound examination of the hands and wrists (the first sites interested at the onset) showed evidence of arthritis, with distention of the joint capsule and synovial hyperplasia in the wrist joints on bilateral sides, and joint effusion at several interphalangeal joints. An X-ray of the hands and wrists was normal. CT scan of the thorax and abdominal ultrasound were performed to rule out generalized lymphadenopathies.

Therefore, a diagnosis of polyarticular, RF-positive juvenile idiopathic arthritis (JIA) was formulated. A complete ophthalmological evaluation with a slit lamp examination excluded the presence of uveitis.

The girl received a course of non-steroidal anti-inflammatory drug (NSAID) therapy with naproxen for four weeks, with no clinical benefit. Concomitant medications (methimazole and LUM-IVA) were not discontinued. In particular, a rheumatological side effect of methimazole treatment, the so-called “antithyroid arthritis syndrome”, was ruled out due to the long latency between methimazole introduction and the onset of arthritis and the presence of the rheumatoid factor and anti-CCP antibodies [3].

Due to the persistence of severe joint disease activity, according to current guidelines [4], a disease-modifying anti-rheumatic drug (DMARD) was required.

The efficacy of methotrexate (MTX) in polyarticular JIA has been demonstrated and it is recommended as a first-line DMARD [4]. Nevertheless, MTX is potentially associated with liver toxicity and its co-administration with other hepatotoxic drugs, such as CFTR modulators, might confer a significant risk of liver injury. For these reasons, MTX is assumed to be contraindicated in patients receiving CFTR modulators. Therefore, one month after the JIA diagnosis, we decided to start treatment with an anti-TNFα agent, etanercept (ETN), at a dosage of 50 mg once a week subcutaneously, following negative screening for tuberculosis, hepatitis B and C. In the meanwhile, therapy elexacaftor (ELX)/tezacaftor (TEZ)/IVA became available on-label in Italy, not only in adults but even in CF patients over the age of 12 years, so treatment with ELX/TEZ/IVA treatment was added 1 month after the first administration of ETN.

After 6 weeks of treatment with ETN, a significant clinical improvement was shown, and after 16 weeks she achieved complete clinical remission of the articular disease. We also observed an improvement in FEV1 (90% before treatment; 100% after treatment), BMI (18.59 before treatment; 19.37 after treatment) and the sweat test (15 mEq/L) after 3 months of therapy with ELX/TEZ/IVA, according to previous data of our patients [5]. Etanercept was continued for 12 months and was still ongoing at our last follow-up. No adverse events occurred, in particular, no infectious episodes or respiratory flare-ups were reported.

## 3. Discussion

Nowadays a precise definition of CF-related arthropathies is missing.

In clinical practice, these patients might experience recurrent monoarthritis or symmetric polyarthritis, often painful and with sudden onset, lasting from 5 to 7 days. This condition involves more frequently the knees, followed in order of frequency by ankles, wrists, hips, shoulders, elbows and the proximal interphalangeal joints [6]. Episodic arthritis usually responds to NSAID therapies or aspirin, although some patients require oral or intraarticular glucocorticoids [1]. In case of recurrence over several years, chronic destructive arthritis might lead to a loss of function of the affected joint [7].

HPOA is the second major rheumatic complication of CF, with a median age of onset of 20 years, and less frequent onset in childhood.

The clinical expression of HPOA has no specific features in CF patients, varying from symmetric polyarthritis with pain and effusions in the knees, wrists and ankles, to the less common synovitis of the hand joints. Radiographs may show a symmetrical periosteal new bone formation at the distal ends of the tibiae, radii, fibulae and ulnae and, occasionally, joint effusions which may be large.

These patients can present a more severe lung disease, a worse pulmonary function than control CF patients (mean FEV~39% in HPOA and 61% in episodic arthritis), and they have a worse prognosis. Furthermore, a characteristic association between the relapse of articular symptoms and infections is reported [6,8]. Thus, optimization of CF management and strict prevention of infection is of utmost importance to prevent pulmonary hypertrophic osteo-arthropathy, while NSAID can be useful to reduce pain and shorten the length of these episodes [1].

Arthritis in CF has also been reported in association with a variety of drugs, especially with some quinolones and cimetidine.

However, some patients might present persistent inflammatory arthritis with a clinical phenotype, resembling chronic inflammatory arthropathies such as rheumatoid arthritis or JIA.

RA showed an association with CF, and before the widespread adoption of new-born screening, several reports were published on RA diagnosis preceding the CF diagnosis even by several years. Furthermore, the association between RA and bronchiectasis suggests that RA may occur at an increased rate in CF patients [1].

In these patients, DMARDs are usually adopted as first-line agents, and TNF-α inhibitors have been used in case of a lack of response to conventional DMARDs.

JIA is one of the most common chronic diseases of childhood. This definition includes several forms of inflammatory arthritis of unknown etiology with onset prior to age 16 years and a minimum of 6 weeks duration, following the exclusion of other known causes of synovitis. According to the current International League of Associations for Rheumatology classification criteria, JIA is divided into 7 categories defined by the number of joints involved, presence or absence of extraarticular manifestations and presence or absence of additional markers including rheumatoid factor (RF) and HLA-B27 [4].

In particular, polyarticular JIA, which is the form of chronic arthritis that affected our patient, includes children with JIA and polyarthritis (≥5 joints ever involved); given their heterogeneity, these patients were categorized into treatment groups using combinations of the following categories: (1) presence or absence of risk factors for disease severity and potentially a more refractory disease course, and (2) low disease activity versus moderate/high disease activity [4].

Risk factors considered are the presence of one or more of the following conditions: positive RF, positive anti-CCP antibodies or joint damage. Disease activity is assessed by the Juvenile Arthritis Disease Activity Score (JADAS), and considering the clinical JADAS based upon 10 joints (cJADAS-10) a cut-off of ≤2.5 versus > 2.5 is used to define low versus high/moderate disease activity [4].

In these patients, treatment with a DMARD is required, and MTX is recommended as the first line DMARD, according to current guidelines, with the possibility to add or change therapy, using biological therapies, if there is no or minimal response to MTX after 6–8 weeks. DMARD is recommended over biologic without and with risk factors, although initial biologic therapy may be appropriate for some patients with risk factors and involvement of high-risk joints, high disease activity and/or those judged by their physician to be at high risk of disabling joint damage [4].

However, the therapeutic challenges of treating patients with CF and chronic arthritis, such as JIA, are underestimated. In fact, medications used for treating JIA have an immunomodulatory effect; therefore, these drugs might theoretically increase the infectious risk, such as mycobacterial infection, in children with CF who are receiving these agents [9]. Several drugs may interact with ELX/TEZ/IVA, such as antifungal medicines and some antibiotics. Nevertheless, little is known about other drugs used for CF-related conditions such as rheumatological or endocrinological diseases.

Up to date, this is the first case evaluating the effectiveness and safety of ELX/TEZ/IVA in a CF patient, concomitant receiving anti-TNFα and methimazole. Etanercept has already been used in patients with CF and arthritis. In particular, Visser et al. [10] described the case of a 17-year-old female CF patient affected by rheumatoid arthritis (RA), and Aldensten et al. [11] described the case of a 46-year-old female CF patient suffering from chronic episodic arthritis; both were treated successfully with TNF-α inhibitor ETN. However, there are no reports of patients with CF and arthritis treated at the same time with ETN, ELX/TEZ/IVA and methimazole. In our patient we hypothesized that the clinical improvement could have been related to the CFTR modulator drug more than to the anti-TNF alpha However, we ruled out this hypothesis as the patient was under treatment with LUM-IVA therapy for three years at the time of arthritis onset and the articular manifestations subsided after only six weeks of treatment with ETN. This latency between the first administration of ETN and the clinical improvement is in line with that observed in clinical practice in patients with isolated JIA. Moreover, the treatment with ELX/TEZ/IVA was initiated after one month of treatment with ETN; therefore, the interval between modification in treatment with CFTR modulating therapy and the clinical benefit on the articular symptoms is too short.

This report seems to reassure regarding the potential side effects of these associations.

In addition, another aspect to consider is related to the ability of anti-TNF drugs to improve systemic inflammation and therefore potentially pulmonary inflammation in patients with CF. According to Visser et al. [10] excessive neutrophilic airway inflammation in CF is responsible for lung damage and a decline in FEV1; therefore, TNF-α, increasing the neutrophilic inflammatory response within the CF lung, represents a logical therapeutic target. However, this class of drugs is known to be associated with an increased risk of infections in healthy patients, which is of particular concern in the CF population. Nevertheless, there are two prior case reports of TNF-α antagonist use in CF, with no report of infective complications in either patient, as was with our patient. In consideration of this, we cannot, therefore, exclude that ETN, in association with specific therapy for CF with ELX/TEZ/IVA, may have contributed to the improvement of our patient’s pulmonary status.

## 4. Conclusions

Our experience suggests that anti-TNFα is an effective option in CF patients affected by JIA and is safe even for children receiving a triple *CFTR* modulator. Moreover, this report seems to reassure regarding the potential side effects of these associations. Further studies in a larger cohort are needed to evaluate the role of this synergic treatment on the quality of life and the long-term management of cystic-related arthropathies.

## Data Availability

Not applicable.

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
