# Peer review of "Concomitant Use of Elexacaftor/Tezacaftor/Ivacaftor and Etanercept in a Cystic Fibrosis Patient with Juvenile Idiopathic Arthritis"

_jcm, 2023, doi:10.3390/jcm12051730_

Round 1

Reviewer 1 Report

In the present report, authors describe an interesting pediatric case of a patient with cystic fibrosis (CF) presenting with RF-positive juvenile idiopathic arthritis, not responding to NSAID. They report clinical improvement after treatment with anti-TNFalpha with concomitant use of CFTR modulator elexacaftor/tezacaftor/ivacaftor, as well as an excellent tolerance of both treatments.

I have some questions for the authors regarding the case report.

-       Was the basedow disease associated with auto-antibodies (TRACK, anti-TPO, anti-thyroglobulin)? Was the patient still receiving methimazole when she developed arthritis? If so, have you considered a possible rheumatologic side-effect of methimazole since arthritis have been associated with this treatment?

-       Regarding the diagnosis of RF-Juvenile idiopathic arthritis itself:

o   Do you confirm that there were no skin involvement?

o   Did you rule out lymph node involvement with CT scan?

o   Could you provide a description of joint radiologic exams (X-ray ? ultrasound ?)

-       Regarding JIA treatment:

o   Etanercept has already been used in patients with CF and arthritis. (Aldensten et al, DOI 10.3109/03009742.2015.1122834, Visser et al., DOI 10.1007/s00408-012-9393-9). Could the authors discuss these papers ?

o   How long was the etanercept continued ?

-       Regarding CFTR modulators:

o   Was LUM/IVA interrupted before the onset of the arthritis?

o   When exactly was ELX/TEZ/IVA started regarding the onset of the arthritis?

o   Have you consider the possibility that the clinical improvement was related to the CFTR modulating therapy more than to the anti-TNFalpha? Please discuss this.

-       Could the authors be more explicit regarding the term “biochemical remission” ?

-       Discussion:

o   Visser et al. suggest etanercept could improve systemic inflammation and thus improve pulmonary exacerbation. Please discuss this.

o   Regarding interaction between ELX/TEZ/IVA : did you monitor blood levels of the drugs while concomitant use with etanercept ?

Minor comments

-       Ethics : Did you obtain the authorization from the patient/her legal tutor to publish the case ?

-       There appeared to be a fex typos that may benefit from a careful re-read (eg : line 45)

Author Response

We thank you for the careful revision of our manuscript entitled “Concomitant use of Elexacaftor/Tezacaftor/Ivacaftor and Etanercept in a Cystic Fibrosis patient with juvenile idiopathic arthritis ”.

  1. Was the basedow disease associated with autoantibodies (TRACK, anti-TPO, anti-thyroglobulin)? If so, have you considered a possible rheumatologic side-effect of methimazole since arthritis have been associated with this treatment?
    1. Reply: we added the autoimmune profile of the patient (“Autoimmune profile showed presence of anti TSH-receptor auto-antibodies, anti-thyroid peroxidase auto-antibodies and antithyroglobulin auto-antibodies.”) at line 47, pg 2. We considered a possible rheumatologic side-effect of methimazole, but in particular, a rheumatological side effect of Methimazole treatment, the so-called “antithyroid arthritis syndrome”, was ruled out due to the long latency between methimazole introduction and the onset of arthritis and the presence of the rheumatoid factor and anti-CCP antibodies. (Takaya K, Kimura N, Hiyoshi T. Antithyroid Arthritis Syndrome: A Case Report and Review of the Literature. Intern Med. 2016;55(24):3627-3633. doi: 10.2169/internalmedicine.55.7379. Epub 2016 Dec 15. PMID: 27980264; PMCID: PMC5283964.)
    2. (See line 47, page 2)

  1. Did you rule out lymph node involvement with CT scan?

Reply: lymph node involvement was excluded by CT scan of the thoracic region only. In order to reduce x-ray cumulative dose in a fertile girl, we performed an echography of the abdomen and clinical examination of other districts (see line 67, page 2)

  1. Could you provide a description of joint radiologic exams (X-ray? ultrasound?)

Reply: the ultrasound and x-ray examination were limitated to sites involved at JIA onset, and we proceeded to insert them (see line 64, page 2)

  1. Regarding interaction between ELX/TEZ/IVA: did you monitor blood levels of the drugs while concomitant use with etanercept?

Reply: we did not monitor blood levels of the drugs as only few facilities have availability of the test worldwide and it is not routinely used in clinical practice.

  1. Have you considered the possibility that the clinical improvement was related to the CFTR modulating therapy more than to the anti-TNFalpha?

Reply: we thank the reviewer for his valuable contribution. Indeed, in our patient we hypotized that the clinical improvement could had been related to the CFTR modulating therapy more than to the anti-TNFalpha. However, we ruled out this hypothesis as the patient was under treatment with lumacaftor-IVA therapy for 3 years at the time of arthritis onset and the articular manifestations subsided after only six weeks of treatment with Etanercept. This latency between first administration of the etanercept and clinical improvement is in line with that observed in clinical practice in patients with JIA isolated. Moreover, the treatment with ELX/TEZ/IVA was initiated after one month of treatment with etanercept, so it is too short the interval between modification in treatment with CFTR modulating therapy and the clinical benefit on the articular symptoms (see line 161, page 4)

  1. Ethics: Did you obtain the authorization from the patient/her legal tutor to publish the case?

Reply: the authorization was obtained, and it is reported at line 191 page 4.

Reviewer 2 Report

In the manuscript “Concomitant use of Elexacaftor/Tezacaftor/Ivacaftor and Etanercept in a Cystic Fibrosis patient with juvenile idiopathic arthritis”, Immacolata and collaborators described a case report of an adolescent with CF that also developed idiopathic arthritis. They follow for 12 months the effects of ETI therapy in combination with Etanercept. This is important due to the potential hepatotoxic effects in drug-drug interaction, particularly as it has been reported in several cases of ETI therapy, despite the significant improvement in lung function.

Specific comments:

1.     There are some typos: wew (line 45), afte (line 79).

2.     Although the acronym for juvenile idiopathic arthritis (JIA) is reported later on in the main text, the authors used it without providing previous information in the Abstract (line 25). Because it is used only once in the Abstract, I recommend replacing it with the full description there.

3.     (line 45) “No related CF disease wew found.” – This sentence seems ambiguous. Please rephrase.

4.     (line 51) “clinical outcomes” – Please specify: FEV1, BMI (?)

5.     (line 80) “(100% vs. 90%), BMI (19.37 vs. 18.59)” – I believe these data should be presented in the opposite direction (before vs. after); otherwise, it seems like she had a worsening instead of improving clinical outcomes.

Author Response

We thanks the reviewer for the valuables comments. 

Here you can find a point-to-point reply:

  1. There are some typos: wew (line 45), afte (line 79).

Thanks, we modified the typos accordingly to suggestion.

  1. Although the acronym for juvenile idiopathic arthritis (JIA) is reported later on in the main text, the authors used it without providing previous information in the Abstract (line 25). Because it is used only once in the Abstract, I recommend replacing it with the full description there.

 We modified the abstract accordingly to the suggestion.

  1. (Line 45) “No related CF disease wew found.” – This sentence seems ambiguous. Please rephrase.

We thank the reviewer, and we rephrase the paragraph in the following way “No related CF diseases were found such as hypochloremic metabolic alkalosis, acute pancreatitis or CF related diabetes” (pg 2, line 45).

  1. (Line 51) “clinical outcomes” – Please specify: FEV1, BMI (?)

We specified the outcome as requested (pg 2, line 54).

  1. (Line 80) “(100% vs. 90%), BMI (19.37 vs. 18.59)” – I believe these data should be presented in the opposite direction (before vs. after); otherwise, it seems like she had a worsening instead of improving clinical outcomes.

We specify in the text “FEV1 (90% before treatment; 100% after treatment), BMI (18.59 before treatment; 19.37 after treatment)” to avoid confusion for the readers (pg 3, line 94).

Round 2

Reviewer 1 Report

I thank the authors for their response to the comments.

The discussion section has been greatly modified compared to the first draft of the article.

I would suggest the authors to :

1) Shorten a little bit the part of the discussion dealing with the joint manifestations in CF and focus on juvenile idiopathic arthritis

2) have the discussion proofed by an english-native speaking reader

3) Clearly indicate the conclusion of their report

Author Response

Dear Editor,

We thank you for the careful revision of our manuscript entitled “Concomitant use of Elexacaftor/Tezacaftor/Ivacaftor and Etanercept in a Cystic Fibrosis patient with juvenile idiopathic arthritis”.

You will find below the point-to-point responses to the reviewers’ comments.

1) Shorten a little bit the part of the discussion dealing with the joint manifestations in CF and focus on juvenile idiopathic arthritis.

Reply: we have reduced the part of the discussion relative to CF arthropaties and we have inserted a focus on the JIA, on the polyarticular form which is the one that interested our patient.

2) Have the discussion proofed by an english-native speaking reader

Reply: the article was revised by a native language speaker, as requested.

3) Clearly indicate the conclusion of their report

Reply: we have added a new section on conclusions.

Kind Regards,

Dr. Sara Immacolata Orsini and Dr. Edoardo Marrani.

Reviewer 2 Report

The authors has responded to all my concerns and I have no further questions.

Author Response

Dear Editor,

We thank you for the careful revision of our manuscript entitled “Concomitant use of Elexacaftor/Tezacaftor/Ivacaftor and Etanercept in a Cystic Fibrosis patient with juvenile idiopathic arthritis ”.

You will find below the point-to-point responses to the reviewers’ comments.

1) Shorten a little bit the part of the discussion dealing with the joint manifestations in CF and focus on juvenile idiopathic arthritis.

Reply: we have reduced the part of the discussion relative to CF arthropaties and we have inserted a focus on the JIA, on the polyarticular form which is the one that interested our patient.

2) Have the discussion proofed by an english-native speaking reader.

Reply: the article was revised by a native language speaker, as requested.

3) Clearly indicate the conclusion of their report.

Reply: we have added a new section on conclusions.

Kind Regards,

Dr. Sara Immacolata Orsini and Dr. Edoardo Marrani.